

# Enabling personalized smart tourism with location-based social networks

Yuqi Shen[1], Yuhan Wu[2], Jingbo Song[3], Xiangjie Kong[1] and Giovanni Pau[4]

[1] College of Computer Science and Technology, Zhejiang University of Technology, Hangzhou, China
[2] School of Software, Dalian University of Technology, Dalian, China
[3] School of Arts, Tourism College of Zhejiang, Hangzhou, China
[4] Faculty of Engineering and Architecture, Kore University of Enna, Enna, Italy

## ABSTRACT

With the rapid advance of mobile internet, communication technology and the Internet of Things (IoT), the tourism industry is undergoing unprecedented transformation. Smart tourism offers users personalized and customized services for travel planning and recommendations. Location-based social networks (LBSNs) play a crucial role in smart tourism industry by providing abundant data sources through their social networking attributes. However, applying LBSNs to smart tourism is a challenge due to the need to deal with complex multi-source information modeling and tourism data sparsity. In this article, to fully harness the potential of LBSNs using deep learning technologies, we propose an knowledge-driven personalized recommendation method for smart tourism. Representation learning techniques can effectively modeling the contextual information (*e.g.*, time, space, and semantics) in LBSNs, while the data augmentation strategy of contrastive learning techniques can explore user personalized travel behaviors and alleviate data sparsity. To demonstrate the effectiveness of the proposed approach, we conducted a case study on trip recommendation. Furthermore, the patterns of human mobility are revealed by exploring the effect of contextual data and tourist potential preferences.

# INTRODUCTION

Benefitting from the rapid development of information and communication technologies, the emergence of smart cities has transformed various aspects of people's lives. By integrating intelligent technologies into urban infrastructure, smart cities achieve intelligent, efficient, and sustainable development goals. Smart tourism is a significant practical application within the context of smart cities, and its components are shown in Fig. 1. It aims to enhance the overall travel experience by offering intelligent services that optimize users' travel arrangements (*Johnson, 2023*; *Wang et al., 2020b*). As a key component of the smart tourism, personalized recommendation systems utilize technologies such as machine learning and data mining to conduct in-depth analysis of travelers' personal interests, travel preferences, and past behaviors. This enables service providers to offer personalized travel planning recommendations and customized services.

Corresponding author
Xiangjie Kong, xjkong@ieee.org

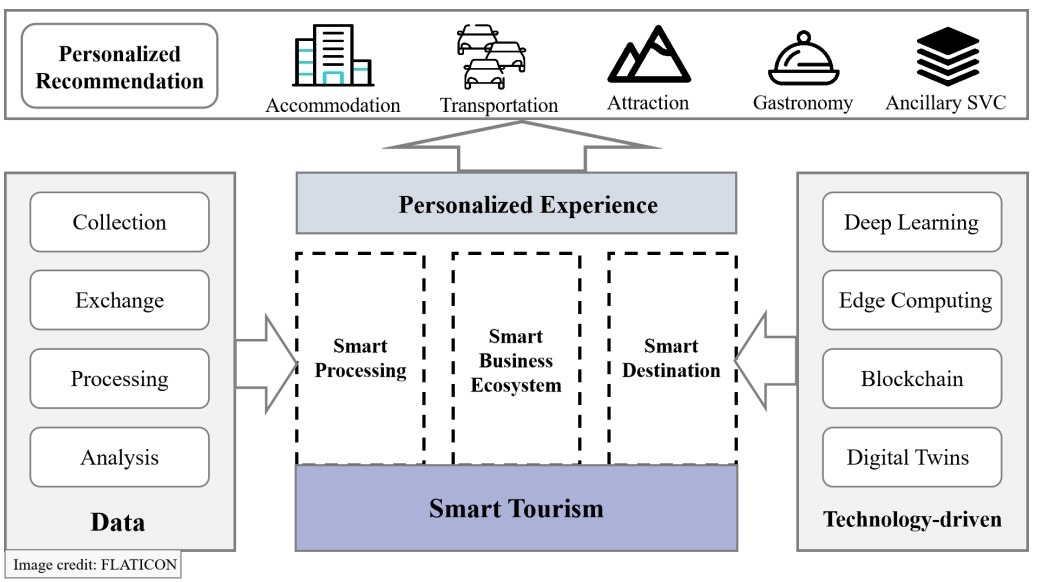

**Figure 1   Components of smart tourism.** Image credit: Flaticon.

The system not only enhances traveler satisfaction and experience quality but also helps the tourism industry improve competitiveness and expand market share.

To better meet the personalized needs of travelers, personalized tourism recommendation systems are exploring the integration with location-based social networks (LBSNs). Unlike traditional social networks, LBSNs not only facilitate connections between users but also capture and exchange users' spatial trajectories and activity information (*Zhou et al., 2023*). The LBSNs enable users to share real-time check-in records with geolocation information in various locations. Such locations that meet users' needs in the real physical world are referred to as points of interest (POI), such as restaurants and entertainment venues. For recommendation systems, the social networking features of LBSNs facilitate users to share travel experiences, recommend attractions, exchange opinions, and provide feedback. These user-generated content and social interactions serve as valuable data sources (*Canturk et al., 2023*). Additionally, LBSNs can provide temporal and spatial contextual information for recommendation systems. By gaining insights into the user's current geographical location and the surrounding environment, personalized recommendation systems can effectively cater to the real-time needs of travelers.

However, applying LBSNs for smart tourism is still challenging. On one hand, users utilizing LBSNs exhibit diverse interests, preferences, and behavioral patterns, which can evolve over time. Effectively modeling user mobility patterns and dynamic preferences is a challenge. Human mobilization, although it is a kind of human behaviour, implies the attributes of the specific spaces where mobility happens (*Wang et al., 2020a*). For example, population exposures in urban greenery were highly correlated with land use distribution and the dynamics of human mobility. On a worldwide scale, the tourism hotspots in the US and EU cities are identified based on geotagged photographs, which

in turn discovers the power-law distribution of the attractiveness of tourist attractions (*Paldino et al., 2015*). On the other hand, personalized recommendation systems encounter data sparsity issues, particularly when integrating multiple data sources. The inherent characteristics and limitations of individual data sources can result in the inclusion of only specific users or travel resources, thereby presenting challenges for recommendation systems in acquiring comprehensive information. Although a number of efforts have been conducted, the discussion about novel solutions to existing challenges of combining smart recommendation systems with LBSNs is missing in other articles.

In this article, we summarize existing efforts and present our view on future directions of smart tourism. We attempt to provide a state-of-the-art article that focuses on existing issues from the perspective of integrating LBSNs, including the dynamics of user preferences and data sparsity. The main contributions of this article are outlined as follows:

1. We perform a case study that combines representation learning techniques and contrastive learning techniques to comprehensively explore the user personalized travel patterns in trip recommendation. The aim is to address the challenges by knowledge-driven personalized recommendation method in integrating smart travel recommendation systems with LBSNs and improve user personalized travel experience.

2. We propose a Spatio-Temporal Contrastive Learning method for POI sequence recommendation that based on contrastive learning and attention mechanisms. Furthermore, we devise four data augmentation techniques aimed at emulating human mobility patterns and mitigating data sparsity.

3. In two real datasets, the effectiveness and impact of different factors in user behavior on recommendation effectiveness is discussed. We further point out the promising prospects for future research on personalized smart tourism.

The remainder of this article is organized as follows. In Section 'Related Work', we systematically review the relevant works on personalized smart tourism and POI recommendation. Our work is focused on Sections 'Methods and Experiments', where we present a case study of smart tourism and offer an extensive discussion on the design principles underlying each component and experimental results of our method on two real datasets. Furthermore, Section 'Opportunities for Personalized Smart Tourism' describe future research directions for smart tourism. Finally, we conclude our work in Section 'Conclusion'.

## RELATED WORK

In this section, we systematically review the relevant works on personalized smart tourism and POI recommendation.

### Personalized smart tourism

With the rapid advancements in mobile internet and intelligent technologies, including artificial intelligence, big data analysis, machine learning, and recommendation algorithms, travel behaviors have undergone significant transformations. There is an increasing trend among users to utilize smart phones and mobile applications for accessing travel

information, planning itineraries, and booking services. This intelligent travel approach serves as the technological foundation for personalized tourism.

Traditional travel planning processes typically require travelers to invest a substantial amount of time and effort in searching for information, organizing itineraries, and dealing with challenges related to information overload and decision-making (*Wang & Wang, 2023*; *Xia et al., 2018*). However, traditional travel recommendation systems often rely on generic strategies or static travel guides, lacking the ability to offer personalized travel recommendations tailored to users' interests, preferences, and time constraints.

Personalized smart tourism encompasses the use of advanced technologies and intelligent algorithms (*e.g.*, next POI recommendation, trip recommendation) to aid users in efficiently filtering and presenting the most pertinent and valuable information, thereby alleviating the cognitive burden associated with information retrieval and filtering (*Kong et al., 2019*) and provide valuable market insights and user insights to tourism enterprises (*Chen et al., 2020*).

## POI recommendation

In contrast to conventional POI recommendation tasks, the POI sequence recommendation problem entails crafting a sequence of POIs adhering to particular spatiotemporal restrictions. Algorithms for POI sequence recommendation fall into two main categories: statistical-based and deep learning-based methods. Statistical-based approaches often draw inspiration from the orienteering problem (OP) and leverage heuristic techniques to optimize the accumulated scores within predefined constraints. These methods integrate specific query constraints and historical data, such as POI popularity and user preferences, to generate trajectory sequences. For example, *Taylor, Lim & Chan (2018)* use Integer Linear Programming (ILP) to recommend a sequence of places, considering factors like starting and ending points, time intervals, duration at each point, and popularity. *Wei, Zheng & Peng (2012)* introduce a collective knowledge-based framework for inferring routes, which extracts spatiotemporal characteristics from uncertain trajectories within predefined location and time constraints. They build a routable graph through a mutually reinforcing approach and employ a routing algorithm to produce top-k popular routes. *Zheng & Xie (2011)* initially capture the historical geographic locations of multiple users using a Tree-based Hierarchical Graph (TBHG) modeling technique. Yet, these methods either depend on local transition distributions, overlook long-term dependencies among POIs, or focus solely on time constraints. Consequently, relying solely on statistical approaches fails to deliver personalized recommendations or grasp the authentic check-in patterns of users.

Recently, numerous studies have turned to deep learning techniques to handle the intricate relationships and features present in data, tackling tasks related to POI sequences by leveraging sequence or location embeddings. Notably, recurrent neural networks (RNNs), long short-term memory (LSTM), and gated recurrent units (GRU) have been utilized to capture semantic connections between user mobility and POI sequences. *Zhou, Mascolo & Zhao (2019)* introduce a holistic deep learning framework that seamlessly merges community and user preferences. They develop a topic memory-enhanced network,

employing neural attention mechanisms and nonlinear methodologies, to amplify both the interpretability and recommendation efficacy of POIs. *Feng et al. (2018)* introduce the DeepMove model, utilizing RNN as its backbone, to account for various factors impacting user mobility *via* a multimodal embedding approach. By employing historical attention mechanisms, they capture diverse periodicities and transition patterns within user flows, facilitating accurate predictions of user mobility (*Feng et al., 2018*). These sequence modeling approaches using RNNs typically capture linear transition patterns between check-in POIs but overlook the sparsity and high dimensionality of user sequences. *Zhou et al. (2018)* present a user trajectory autoencoder model grounded in semi-supervised learning. Their model relies on robust assumptions regarding the distribution of user sequences and effectively encodes semantic information within and across check-in sequences to capture human mobility patterns. *Gao et al. (2021)* introduce DeepTrip, an adversarial neural network model employing a generator and discriminator to produce query and travel representations, enhancing the recommendation of optimal routes. The model delves deeply into contextual POI information and employs an encoder–decoder structure to capture user mobility. *Kong et al. (2024)* extracted temporal dependence through time series decomposition and autocorrelation mechanisms, and extracted spatial dependence through learnable adaptive graph convolution operations. However, these deep learning techniques neglect user diversity, excessively depend on historical data, and struggle to accurately model the intricate and uncertain nature of user check-in behavior. In order to emphasise more clearly the place and importance of this study in the literature, we have added a literature table containing various important columns such as model name, dataset, evaluation, *etc.*, as shown in Table 1.

In this study, we will employ self-supervised contrastive learning and incorporate user travel patterns to simulate users' real travel behavior, thereby enhancing the POI sequence recommendation problem.

## METHODS

To better illustrate the significance of LBSNS and AI-empowered recommendation system for smart tourism, we show a case study on trip recommendation. Trip recommendation is a significant part of smart tourism, which aims to offer tourist a sequential arrangement of POIs considering specific spatio-temporal constraints.

This section introduces the core components of the our method, which involve leveraging representation learning to investigate the contextual information associated in LBSNs and utilizing contrastive learning to mitigate travel data sparsity. The overall framework of the method is illustrated in the Fig. 2.

### Exploring POIs in LBSNs
#### *POI context information modeling*
Location-based recommendation methods not only need to consider user preferences but also take into account the contextual information related to POIs in LBSNs, including

**Table 1 List of related literature.** For each article, we describe the name of the corresponding proposed model, the modules used in the proposed solution, the metrics used for performance evaluation, and the disadvantage of the model. The papers are sorted by year of publication in decreasing order.

| Model | Dataset | Modules | Evaluation | Disadvantage |
|---|---|---|---|---|
| MPGNNFormer (*Kong et al., 2024*) | The real-world bus dataset | GCN, Attention | MAE, RMSE, MAPE | Although adaptive graph convolution is proposed to reduce the computational complexity, Transformer models and graph neural networks usually still have high computational cost when dealing with large-scale graph data. |
| DeepTrip (*Gao et al., 2021*) | Flickr, Foursquare | RNN, GAN | $F_1$ score, pairs-$F_1$ score | There are data sparsity and overfitting problems. |
| NASR+ (*Wang, Wu & Zhao, 2021*) | The Beijing taxi dataset, The Porto taxi dataset, The Beijing bicycle dataset | RNN, GNN, MLP | $F_1$ score, Precision, Recall, Edit Distance | There are data sparsity and high computational complexity problem. |
| TEMN (*Zhou, Mascolo & Zhao, 2019*) | WeChat | Memory Network | HR@k, NDCG@k | This is a complex hybrid model model training and tuning is more difficult and the generalization ability has not been verified. |
| DeepMove (*Feng et al., 2018*) | Foursquare | RNN | Top-1 accuracy | This method focuses mainly on predicting the next position without considering real-time prediction and adaptation in dynamic environments. |
| TULVAE (*Zhou et al., 2018*) | Foursquare | RNN, VAE | ACC@K, macro-P, macro-R, macro-F1 | The TULVAE model, which combines hierarchical trajectory modeling and latent representation, may not be effective enough on small datasets, especially when the number of users is small. |
| LP+M (*Taylor, Lim & Chan, 2018*) | Flickr | ILP | Inclusion of Must-see POIs, Tour Profit, POIs Visited, Utilized Budget | ILP models can be computationally challenging when dealing with large-scale datasets and are not flexible enough to handle user-specific preferences and constraints. |
| RICK (*Wei, Zheng & Peng, 2012*) | Foursquare, The Beijing taxi dataset | A* algorithm | Route score, NDTW, MD | When data points are sparse, the accuracy of route inference may be affected. |

spatial information, time information, semantic information, and sequence information, as shown in Fig. 3. The modeling process is as follows:

1. **Spatial information.** Our goal is to investigate users' underlying geographic preferences, particularly their inclination towards selecting nearby check-in locations when choosing their next destination. To achieve this, We set a geographic threshold of 2 km and established the spatial connections among POIs within this threshold. We construct a POI-geographical graph $G_{vv} = (\mathcal{V} \cup \mathcal{V}, \mathcal{E}_{vv})$, where $\mathcal{V}$ denotes a set of POI, $\mathcal{E}_{vv}$ is the set of edges between POIs.

2. **Time information.** Users may have a preference for visiting certain POI during specific time periods, such as visiting museums in the morning and going shopping in the afternoon. In order to model the periodic patterns in time, we define a POI-time

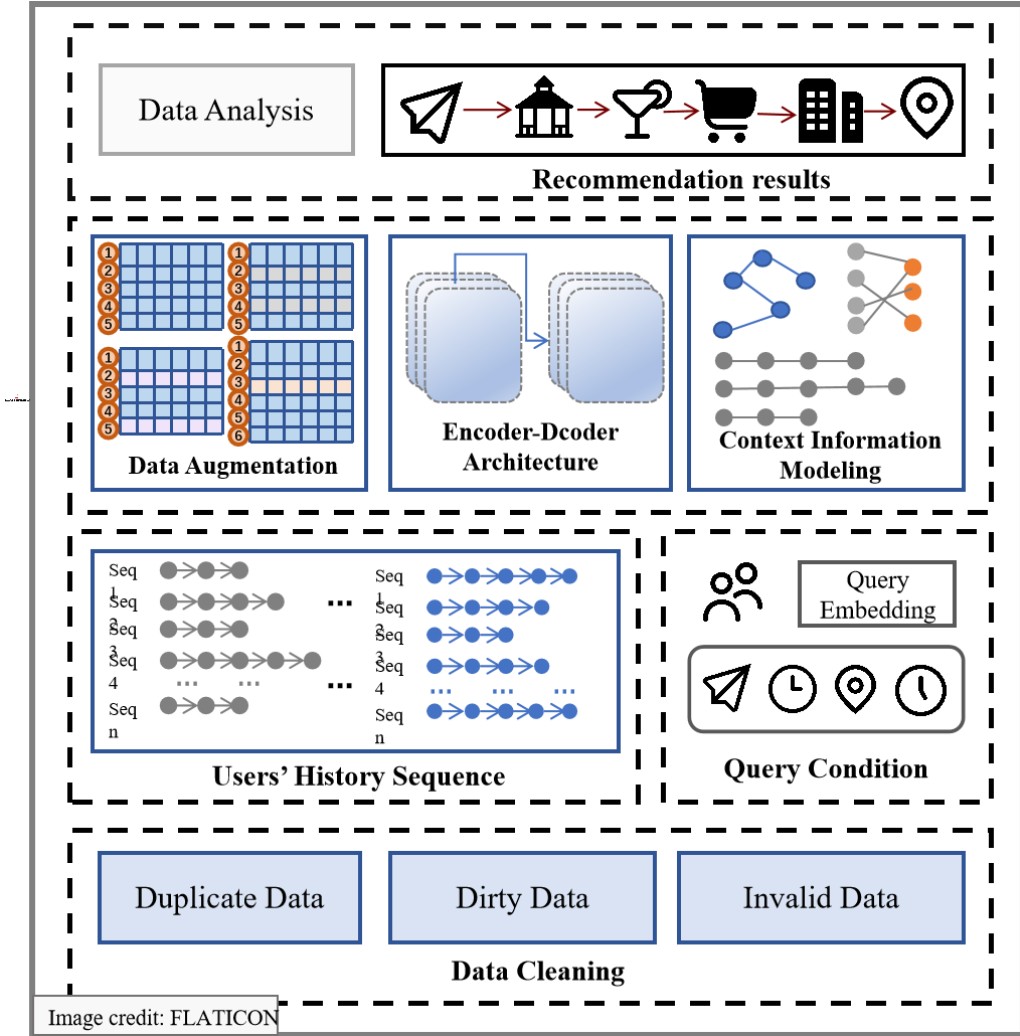

**Figure 2 Framework of LBSNs and AI-empowered travel system for trip recommendation.**

graph $G_{vt} = (\mathcal{V} \cup \mathcal{T}, \mathcal{E}_{vt})$, where $\mathcal{T}$ denotes the set of time-stamp, $\mathcal{E}_{vt}$ is the set of edges between POI and timestamp.

3. **Semantic information.** Our aim is to explore the relationships between POI, specifically by modeling an POI-category graph $G_{vc} = (\mathcal{V} \cup \mathcal{C}, \mathcal{E}_{vc})$, where $\mathcal{C}$ denotes the set of category, $\mathcal{E}_{vc}$ is the set of edges between POI and category.

4. **Sequence information.** Human mobility patterns exhibit strong sequential patterns, where the transition from one checked-in POI to another follows a non-uniform distribution. In order to leverage sequential effect, we employ text analysis methods to construct relationships between POI. Specifically, each POI is treated as a word, each sequence of checked-in POI is treated as a sentence, and all sequences of checked-in POI are treated as a corpus to be represented.

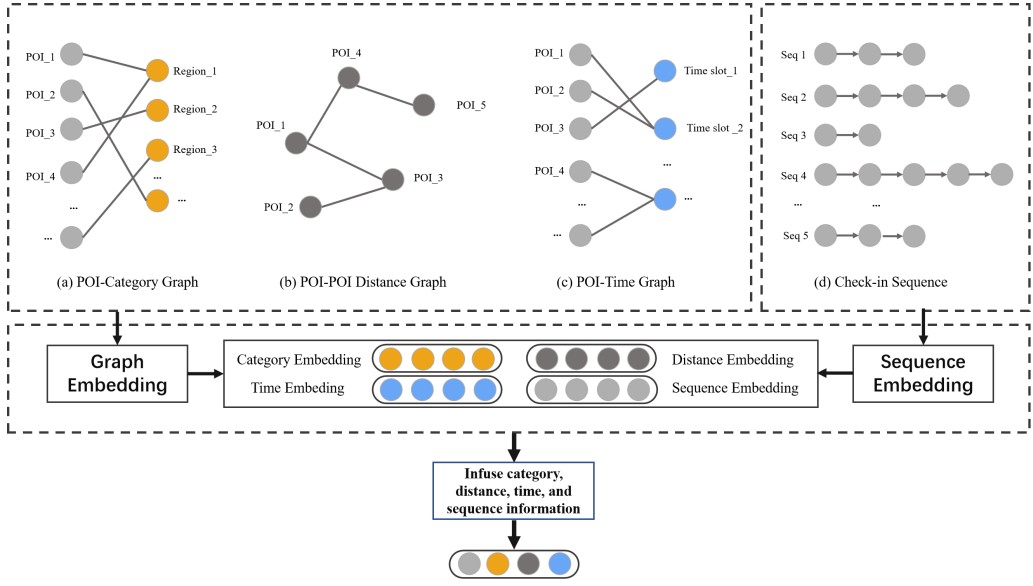

**Figure 3** The exploration of POI context informaiton in LBSNs.

### POI context information embedding

Currently, most graph representation methods primarily focus on node similarity and overlook the similarity among node neighbors. To address these issues, large-scale information network embedding (LINE) technique introduces first-order and second-order proximities to constrain the learning of nodes in a homogeneous graph, thereby preserving both local and global network structures (*Tang et al., 2015*). In this article, we uilitize LINE to embed aboved three graph structure. For a given undirected graph $G = (\mathcal{V}_A \cup \mathcal{V}_B, \mathcal{E})$, where $\mathcal{V}_a$ and $\mathcal{V}_b$ are two different sets of node types and e represents the edges between heterogeneous nodes. We define the conditional probability of a node $v_j$ in the node set $V_B$ being generated by a node $v_i$ in the node set $V_A$ as:

$$p\left(v_j | v_i\right) = \frac{\exp\left(\vec{v}_j^T \cdot \vec{v}_i\right)}{\sum_{v_k \in v_B} \exp\left(\vec{v}_k^T \cdot \vec{v}_i\right)} \tag{1}$$

where $\vec{v}_i$ the representation vector of node $v_i$ and $\vec{v}_j$ is the representation vector of node $v_j$. By preserving the weight $w_{ij}$ of the edge $e_{ij}$, the approximate empirical distribution of the dimension-reduced $p(v_j | v_i)$ is defined:

$$\hat{p}\left(v_j | v_i\right) = \frac{w_{ij}}{d_i} \tag{2}$$

where $d_i$ is the degree of node $v_i$, with $d_i = \sum_j w_{ij}$. The above two formulas define the conditional distribution $p(\cdot | v_i)$ and $\hat{p}(\cdot | v_i)$ of nodes in $\mathcal{V}_B$, and the next step is to define and minimize the objective function:

$$O = \sum_{v_i \in v_A} \lambda_i KL\left(p, (\cdot | v_i), \quad \hat{p}(\cdot | v_i)\right) \tag{3}$$

where $\lambda_i$ is the importance of $v_i$ nodes in the graph. This optimization objective is expensive because it requires traversing the entire set of nodes of the graph. Here, a Negative Sampling strategy is used to sample multiple negative example edges from the noisy space parameterized by the weight coefficients, and the final objective function is as follows:

$$O = \log \sigma \left( \vec{v}_j^T \cdot \vec{v}_i \right) + \sum_{n=1}^{K} E_{v_n \sim P_n(v)} \left[ \log \sigma \left( -\vec{v}_n^T \cdot \vec{v}_i \right) \right] \tag{4}$$

where $K$ is the number of negative sampling edges, $\sigma(x) = \frac{1}{1+e^{-x}}$ is the sigmoid activation function, $P_n(v) \propto d_v^{3/4}$, where $d_v$ is the out-degree of the node.

For sequence information, we use Skip-gram model of Word2vec to map POIs to low-dimensional vector space while preserving the semantics similarity and sequential relationship between POIs. Given an POI, the Skip-gram model provides a series of context POIs ranging from $v_{i-w}$ to $v_{i+w}$, thereby producing the embedding of sequence. Accordingly, the objective function is defined as follows:

$$\mathcal{L} = \frac{1}{|S_u^n|} \sum_{v_i \in S_u^n} \sum_{-w \leq k \leq w, k \neq 0 \; i+k} p(v_{i+k}|v_i)$$

$$p(v_{i+k}|v_i) = \frac{\exp \left( \vec{v}_{i+k}^T \cdot \vec{v}_i \right)}{\sum_{v_j \in \mathcal{V}} \exp \left( \vec{v}_j^T \cdot \vec{v}_i \right)}. \tag{5}$$

After obtaining the geographical distance representation $d$, the timestamp representation $t$, the category representation $c$, and the sequence representation $v$, the representation vectors $d_v$, $t_v$ and $c_v$ of the POIs are extracted from $d$, $t$, and $c$, respectively. These vectors are then concatenated with the sequence representation v to form the context representation vectors $s = [v; d_v; t_v; c_v]$. Hence, the vector s incorporates a variety of complex contextual information in LBSNs.

## Alleviate data sparsity by contrastive learning

Existing supervised models for trip recommendation overly emphasize performance and overlook the potential correlation between contextual data and POI sequence data, resulting in inefficient data representation. To address the above issues, this study draws inspiration from self-supervised learning to enhance POI sequence recommendation. The ability of supervised learning to automatically learn from large amounts of labeled data has led to its wide application in fields such as natural language processing and computer vision. Models represented by SimCLR and MoCo have achieved remarkable results on the ImageNet dataset. The SimCLR model generates positive samples by rotating the image, cropping the image, adding Gaussian noise, and coloring (*Chen et al., 2020*). The MoCo model encodes negative samples using a momentum encoder to increase the number of negative samples, and also employs a sliding average strategy to ensure timeliness (*He et al., 2020*). In our method, a contrastive learning module is designed to pretrain the model and explore different data augmentation strategies to uncover the intrinsic correlations within the data, thereby enriching the self-supervised signals for enhancing data representation and alleviate data sparsity. The data augmentation strategies can be denoted as follows.

1. **Token shuffling.** The purpose of token shuffling is to randomly shuffle the order of token in an input sequence to increase the robustness and generalization performance of the model. When mapping token shuffling strategy to real travel behavior, it can be modeled as the scenario where even if the recommendation method suggests a sequence of POI to the user, the user may change the entire trip due to immediate preferences. For example, the method suggests a sequence of POIs $< A, B, C, D, E, F >$ to a user, but the user may temporarily change it to $< A, C, E, B, D, F >$ due to various factors. This is a result of the diversity of user preferences. To simulate this disruptive behavior, we randomly shuffle the order of POI in the sequence, create a new trip route, and add it to the training set. Define the sequence of POI $[v_i, \ldots, v_{i+r+1}]$ of S to be $[v_i^m, \ldots, v_{i+r+1}^m]$, and $S^R = R(S) = [v_1, v_2, \ldots, v_i^m, v_{i+r-1}^m, \ldots, v_n]$, where $r = \lceil \omega n \rceil$ is the length of the sub-interest sequence, $0 \leq \omega \leq 1$.

2. **Random deletion.** By applying the random deletion strategy to trip recommendation, we simulate the behavior of users temporarily not wanting to visit certain POI, thereby creating a new itinerary view. For example, users may change their trip plans for various reasons (*e.g.*, personal reasons, weather conditions) and decide to skip or cancel visiting the next POI and proceed to the following one. Define the sequence $S^M = M(S) = [v_1^m, v_2^m, \ldots, v_n^m]$, where $v_1^m$ represents that if $v_1$ is selected, it will be erased; otherwise, $v_1^m = v_1$. Define $l = \lceil \mu n \rceil$ as the set of points to be erased, where $l$ is controlled by the hyperparameter $\mu$, with $0 \leq \mu \leq 1$.

3. **Random insertion.** In the check-in dataset, the user's POI check-in sequences are often sparse, making it difficult to fully capture the user's preferences and the relevance of POIs when training with these sequences. To address this issue, the approach of inserting POI can be used to construct expanded check-in sequences. Specifically, we randomly select $k$ different POI indices $\{idx_1, idx_2, \ldots, idx_k\}$ from the POI sequence, where $k = \lceil \alpha n \rceil$, $idx_i \in [1, 2, \ldots, n]$, $\alpha \in [0, 1]$ is the substitution rate. The substituted sequence is $S^s = S(S) = [v_1, v_2, \ldots, \bar{v}_{idx_i}, v_{|S|}]$, where $\bar{v}_{idx_i}$ is the substituted interest point.

4. **Synonyms substitution.** The purpose is to recommend similar and substitutable POI to users in order to discover their additional real interests. Replacing elements in the sequence with highly correlated POIs reduces the information loss of the original sequence, thereby generating high-quality positive pairs. In trip recommendation, recommending similar and substitutable items can help users discover more of their actual interests. Meanwhile, users may have a preference for visiting similar POI within a geographic threshold. Specifically, the ratio of inserted interest points is controlled by $\beta \in [0, 1]$. Randomly select $k$ different POI indices $\{idx_1, idx_2, \ldots, idx_k\}$, where $k = \lceil \beta n \rceil$, $idx_i \in [1, 2, \ldots, n]$. The sequence after insertion is $S^I = I(S) = [v_1, v_2, \ldots, \bar{v}_{idx_i}, v_{idx_i}, \ldots, v_n]$, where $\bar{v}_{idx_i}$ is the POI related to $v_{idx_i}$. The length of the expanded sequence $S^I$ is $k + n$.

During the contrastive learning phase, by implementing the objective of having positive examples of POI close to the sequence and negative examples far from the sequence (*Oord, Li & Vinyals, 2018*), the correlation between POI and sequences is used as an additional signal to improve trip recommendation effectiveness.

For a given set of interest point sequences S, for each interest point sequence $S_i$, two augmentation strategies are randomly selected from the four strategies mentioned above. This results in an augmented sequence of 2N elements $S_1, S_2, \ldots, S_{2i-1}, S_{2i}, \ldots, S_{2N-1}, S_{2N}$, where i 1, 2, ..., N. The pairs $(S_{2i-1}, S_{2i})$ are treated as positive pairs, while the remaining 2(N-1) augmented views serve as negative pairs for each positive pair. After encoding by the model's decoder, each positive pair is represented as $(h_{2i-1}, h_{2i})$. In this paper, we adopt Noise Contrastive Estimation (NCE) as the loss function. The final pre-training loss function is as follows:

$$\mathcal{L} = \log \frac{\exp\left(\text{sim}\left(\bar{h}_{2i-1}, \bar{h}_{2i}\right)\right)}{\exp\left(\text{sim}\left(\bar{h}_{2i-1}, \bar{h}_{2i}\right)\right) + \sum_{k=1, k \neq 2i-1}^{2N} \exp\left(\text{sim}\left(\bar{h}_{2i-1}, \bar{h}_{k}\right)\right)}. \tag{6}$$

**The overall process of framwork**

First, to convert the user's query conditions into vector embeddings, the contextual representations of the POI obtained from Section 'Alleviate data sparsity by contrastive learning' are used to transform the start location $l_s$ and the destination location $l_d$. In addition, the timestamps are divided into 24 h, and the start time $t_s$ and end time $t_d$ are one-hot encoded. After obtaining the embedding representations, we uses concatenation to obtain the query condition $q$:

$$q = LeakyRelu([s(l_s)|t(t_s)|s(l_d)|t(t_d)]W_q + b_q). \tag{7}$$

Second, we use an attention mechanism to fuse the query vector $q$ and the historical sequence vector $S$, and the fused vector contains both historical information and query information. Through the attention mechanism, the model automatically learns which historical visited POIs are most relevant to the current query and then takes the weighted average of the representations of these locations to obtain a new vector $Z$.

Third, We input the fused vector $Z$ into a encoder–decoder architecture and train the neural network using the parameters learned from pre-training, ultimately obtaining the recommendation results (*Vaswani et al., 2017*). The final loss function is defined as follows:

$$\mathcal{L} = \|\widehat{Src} - Target\|. \tag{8}$$

In the above, $\widehat{Src}$ represents the output of the decoder, and *Target* represents the real POI sequence.

# EXPERIMENTS

In this section, a series of experiments and validations is discussed.

## Experiment settings
### Datasets
The dataset used in this article is the Yahoo Flicker Creative Commons 100 Million Dataset (The dataset is available at https://webscope.sandbox.yahoo.com/catalog.php?datatype= i& did=67) (YFCC 100M), which is the world's largest image database since 2014 from Yahoo

**Table 2  Description of two real world datasets.**

| Dataset | #user | #check-in | #poi sequence |
|---|---|---|---|
| YFCC 100M@Osaka | 450 | 7747 | 1115 |
| YFCC 100M@Glasgow | 601 | 11434 | 2227 |

Flicker (*He, Qi & Ramamohanarao, 2019*). It includes 90 million photos and 1 million videos from around the world between 2004 and 2014. All geotagged image information from two cities, Osaka and Glasgow was extracted from the YFCC 100M dataset in this article. We selected the POI data contained in these two cities separately, which comes from the open-source POI dataset. There are more than five types of POI, covering aspects such as amusement, park, historical, religious, entertainment.The data for the two cities are described in the Table 2.

To ensure the quality of the dataset, we first filters out check-in sequences with less than three POIs, as such sequences are insufficient to reflect the user's actual behavior. In addition, the timestamps are standardized to a 24-hour format with an hourly granularity. Furthermore, to ensure the accuracy of the geographical latitude and longitude, only the high-precision Flicker image set is selected.

### Evaluation metrics

To evaluate the model performance, we employ $F_1$ and pairs-$F_1$ as performance metrics, which are widely used in related works. $F_1$ can effectively measure the quality of the recommended sequence of POI. Similarly, pairs-$F_1$ can measure the accuracy and sequence order of every pair of POI in the sequence, regardless of whether they are adjacent or not.

### Baselines

We compare our proposed approach with the following five baseline methods:

- **Markov** (*Chen, Ong & Xie, 2016*). Constructing a POI transition matrix is a commonly used and intuitive approach for trip recommendation.
- **Markov-Rank** (*Chen, Ong & Xie, 2016*). The Markov model combined with POI rank information .
- **POIRank** (*Chen, Ong & Xie, 2016*). POIRank utilizes POI transition matrix and rankSVM with linear kernel to learn rating ranking, and recommends POI sequences based on Markov chains.
- **DeepTrip** (*Gao et al., 2021*). This study employs an encoder–decoder architecture and generative adversarial network to capture user mobility patterns and model the transitional distribution of POI.
- **NASR** (*Wang, Wu & Zhao, 2021*). This work improves the search algorithm of neural networks by learning the cost function of the widely used $A^*$ algorithm.

### Implementation details

We reproduce the benchmarks and implement our method in Tensorflow while methods are accelerated by two NIVDIA RTX 3080Ti GPU. For our proposed method, we set the number of encoder layer and decoder layer as 4. The number of attention heads is 8, and
the dimensionality of POI and model is set as 256 and 128, respectively. In the pre-training stage, the mask proportion of POI in random deletion strategy is set as 0.5. The model are optimized by Adam optimizer with learning rate of 0.1 and the batch size is set as 8 and 16 in the pre-training and fine-tuing stage, respectively.

## Performance comparison

The performance of our proposed model and the eight baselines on two datasets evaluated by $F_1$ and pairs-$F_1$ is shown in Table 3. Our methods achieves the best results in both $F_1$ score and pairs-$F_1$ score, demonstrating the highest accuracy compared to other methods. The Markov-based methods focus more on transitions between POIs but overlook other contextual information, resulting in the poorest performance compared to other methods. On the other hand, the rank-based methods, Markov-Rank and POIRank, capture user mobility by considering both co-occurrence patterns and feature information of POIs, outperforming the Markov-based method. However, due to their reliance on statistical modeling or machine learning methods, these approaches fail to fully exploit users' complex mobility patterns and their short and long-term preferences, leading to inferior performance compared to deep learning models such as DeepTrip and NASR. Although DeepTrip combines latent variables and utilizes adversarial generative neural network structures to capture users' visiting intentions and mobility patterns, its performance is affected by data sparsity and ranks second. Our model exhibits the best performance. This is attributed to its effective fusion of POI contextual features, the design of intuitive augmentation strategies to simulate users' real check-in behavior, and the provision of self-supervised signals for adequately modeling users' complex demands.

Next, we analyze the results of recommendation visualization using Osaka as the validation city for trip recommendation, as shown in Fig. 4. Users provide the starting and ending locations of POIs, along with the corresponding time information. The figure displays the recommendation results of three baseline methods: Markov, Markov Ranking, DeepTrip, as well as our proposed method. From the analysis of the figure, it can be observed that the Markov-based methods can only form a short sequence of POIs. This is because this method focus on capturing local features and transition relationships, only using the last POI in the sequence to recommend the next one. The Markov algorithm will result in redundant POIs, as shown in the figure. By introducing POI ranking information, the Markov-Rank methods can effectively reduce the occurrence of redundant POIs but perform poorly in terms of the global sequence distribution. DeepTrip can capture global sequence patterns and outperforms the previous two methods, but it falls short in dealing with the issue of data sparsity and insufficient modeling of sequence position information, resulting in inferior recommendation performance compared to our method.

## Data augmentation analysis

In this section, we evaluate the effect of various trip augmentation strategies for recommendation performance. We consider five options for each transformation, including aforementioned four strategies and None, resulting in $5 \times 5$ combinations. Especially, None

**Table 3  The recommendation performance comparison on the two datasets.**

| Method | Osaka | | | | Glasgow | | | |
|---|---|---|---|---|---|---|---|---|
| | $F_1$ | Improve | pairs-$F_1$ | Improve | $F_1$ | Improve | pairs-$F_1$ | Improve |
| Markov | 0.634 | 25.1% | 0.385 | 55.1% | 0.687 | 20.2% | 0.427 | 45.6% |
| Markov-Rank | 0.689 | 18.6% | 0.497 | 42.1% | 0.695 | 15.6% | 0.454 | 42.2% |
| POIRank | 0.705 | 16.7% | 0.535 | 37.6% | 0.773 | 6.18% | 0.547 | 30.4% |
| DeepTrip | 0.798 | 5.78% | 0.723 | 15.7% | 0.810 | 1.69% | 0.769 | 2.16% |
| NASR | 0.815 | 3.77% | 0.805 | 6.17% | 0.806 | 2.18% | 0.731 | 6.97% |
| **Our Model** | **0.864** | | **0.865** | | **0.824** | | **0.786** | |

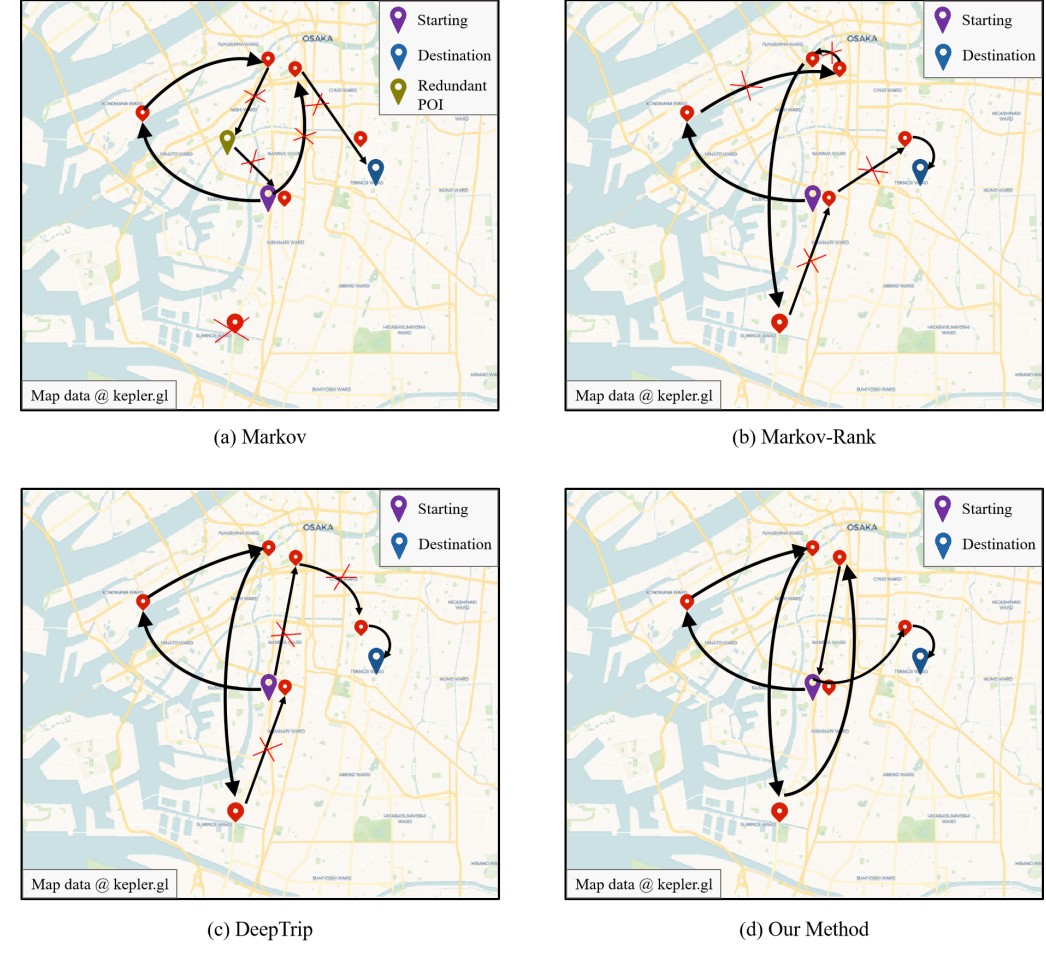

(a) Markov

(b) Markov-Rank

(c) DeepTrip

(d) Our Method

**Figure 4  The visualization of recommendation results.**

means we do nothing and diagonal implies that we employ the same augmentation strategy for a sequences.

The testing results with different couples of augmentation strategy can be found in Fig. 5. We can make the following observations. First, Random Insertion is the most

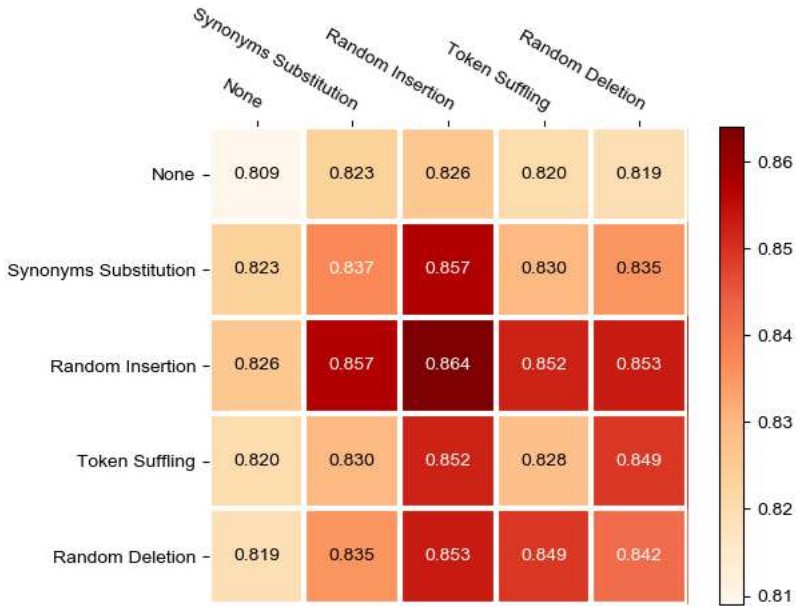

**Figure 5 The visualization of four augmentation stratgies.**

effective strategies, significantly outperforming other augmentation strategies. By solely utilizing the random insertion strategy, we obtained the optimal outcome. Additionally, when integrating random insertion with synonyms subtitution, we achieved the second highest performance. The reason for this phenomenon may be attributed to the fact that in recommendation model, which recommend a series of consecutive POIs, the sequence length is typically short. Therefore, both the random insert strategy and the synonyms subtitution strategy can effectively perturb the correlation between POIs, resulting in augmented sequences with higher confidence, especially with the random insertion strategy. As a result, whether employing only the random insert strategy or a combination of random insertion and synonyms subtitution, the model achieves remarkably high performance. Random deletion performs slightly worse than random insertion but better than other strategies, possibly because it alters the structure of the sequence and generates hard examples.

## Impact of different infactors

In this part, we first verified the impact of individual factors among various factors in exploring POI context representation on the overall performance. Additionally, we examined the influence of the contrastive learning module on the model's effectiveness. Hence, we use W/O to represent the removal of the corresponding factors, and we design several variants of proposed method to verify the impact of each information, as shown in Fig. 6.

After conducting ablation experiments on Osaka datasets, the method that incorporated the context feature module achieved the best performance. This indicates the significance

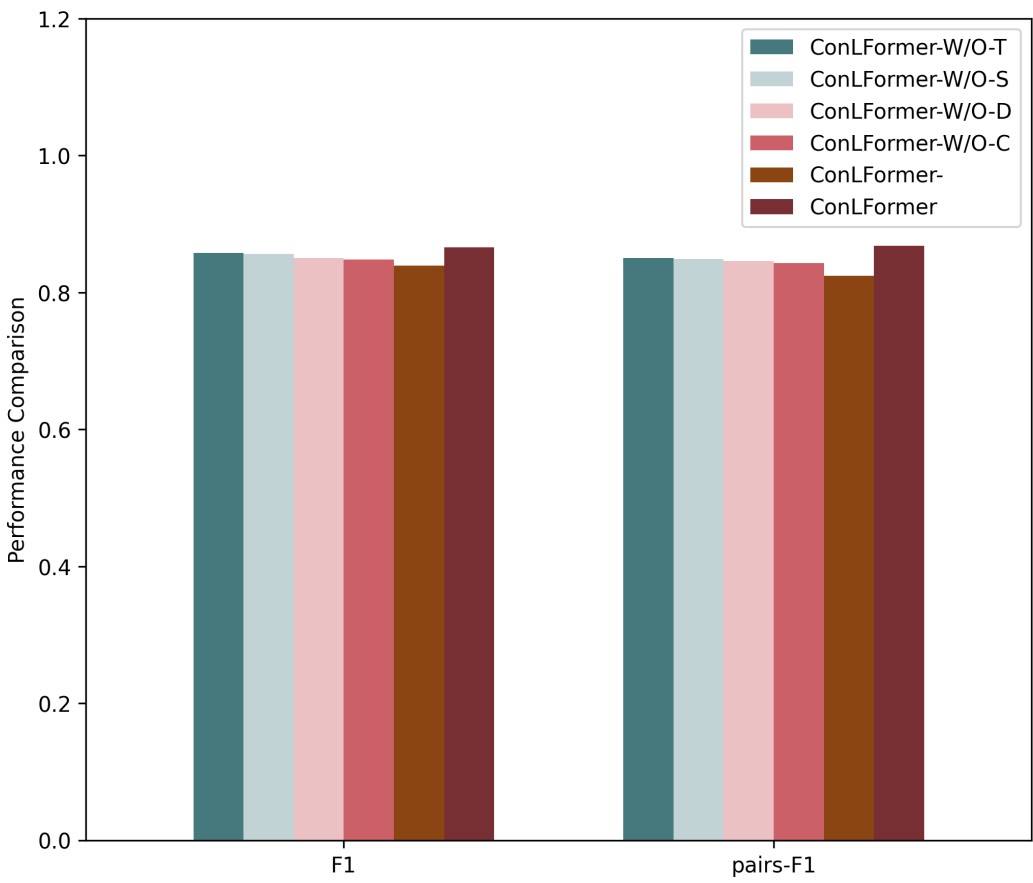

**Figure 6** The comparison of various infactors.

of incorporating POI context information into the model. Compared to relying solely on a single sequence representation or a few representations, a more comprehensive set of information influences user decision-making. In the Osaka dataset, the sampling methods that removed the POI distance representation and POI category representation achieved the first and second worst performances, respectively. Removing the remaining two representations yielded slightly lower performance. This suggests a strong correlation between user check-in patterns in Osaka and time factors, indicating that users may be influenced more by their time periodic preferences when making decisions. This shows that there is a strong correlation between user check-in patterns in Osaka and geographical factors, indicating that users may be more affected by the distance of the check-in point and the POI itself when making a decision.

Additionally, through the exploration of the contrastive learning module on both datasets, we found that this module effectively improved the overall performance of the algorithm in terms of the $F_1$ score and pairs-$F_1$ score. This demonstrates that employing data augmentation strategies can enhance data representation, adequately modeling complex travel patterns and dynamic preferences of users. It indicates that the sparsity of

sequence data and the heterogeneity of user needs are critical factors influencing sequence recommendation.

## Parameters analysis

This section investigates the sensitivity of some key hyperparameters, including (1) the number of layers in the encoder and decoder, $n_{layer}$, (2) the number of heads in the multi-head attention mechanism, $n_{head}$, and (3) the model's embedding dimension, $d_{model}$. We conduct extensive experiments to analyze the importance of these parameters. The chosen parameters for this study are: $n_{layer} = 4$, $n_{head} = 8$, $d_{model} = 128$. As shown in Fig. 7, the model's representational dimension and the number of encoder–decoder layers cause more drastic changes in model performance, with the model dimension being the most significant. The possible reason is that a higher number of parameters means the model can learn and represent the complex relationships within the data more fully. When adjusted, the model has a greater capacity to adapt to a wider range of data characteristics. Model performance is most sensitive to the model's dimension. When adjusting the number of attention heads, the model performance does not change significantly, which may be because the dataset does not have particularly complex dependencies that would lead to a notable performance improvement.

## OPPORTUNITIES FOR PERSONALIZED SMART TOURISM

Through a series of evaluations, the effectiveness of our method in trip recommendation tasks has been demonstrated. However, there are still promising directions for smart toursim, especially trip recommendation. First, for the exploration of LBSNs, additional information such as social and weather data can be incorporated to enrich the contextual data. Furthermore, in modeling user mobility patterns and alleviating data sparsity, exploring more effective data augmentation strategies to capture dynamic user needs, such as transferring some augmentation strategies from the computer vision or natural language processing domains, may be considered. Finally, in method selection, the complexity of the encoder–decoder structure itself can limit its performance when dealing with large-scale data. Simplifying the structural design to achieve lower computational complexity could be explored for smart and personalized recommendation.

## CONCLUSION

This work focuses on the core application of smart tourism, which is personalized smart travel recommendation system. The goal is to provide users with satisfying and customized travel services by integrating with LBSNs. We first described the overview of smart tourism and discussed the main characteristics and significance. To fully explore the potential of LBSNs by deep learning technologies, this article proposed a method that combines POI context presentation learning with contrastive learning to model tourist personalized behavior and alleviate data sparsity. Then, extensive experiments and evaluations on two real LBSNs datasets demonstrate the effectiveness of the our work. Specially, we analyzed the impact of different data augmentation strategies and various contextual information on user personalized behavior modeling for smart recommendation. Finally, considering

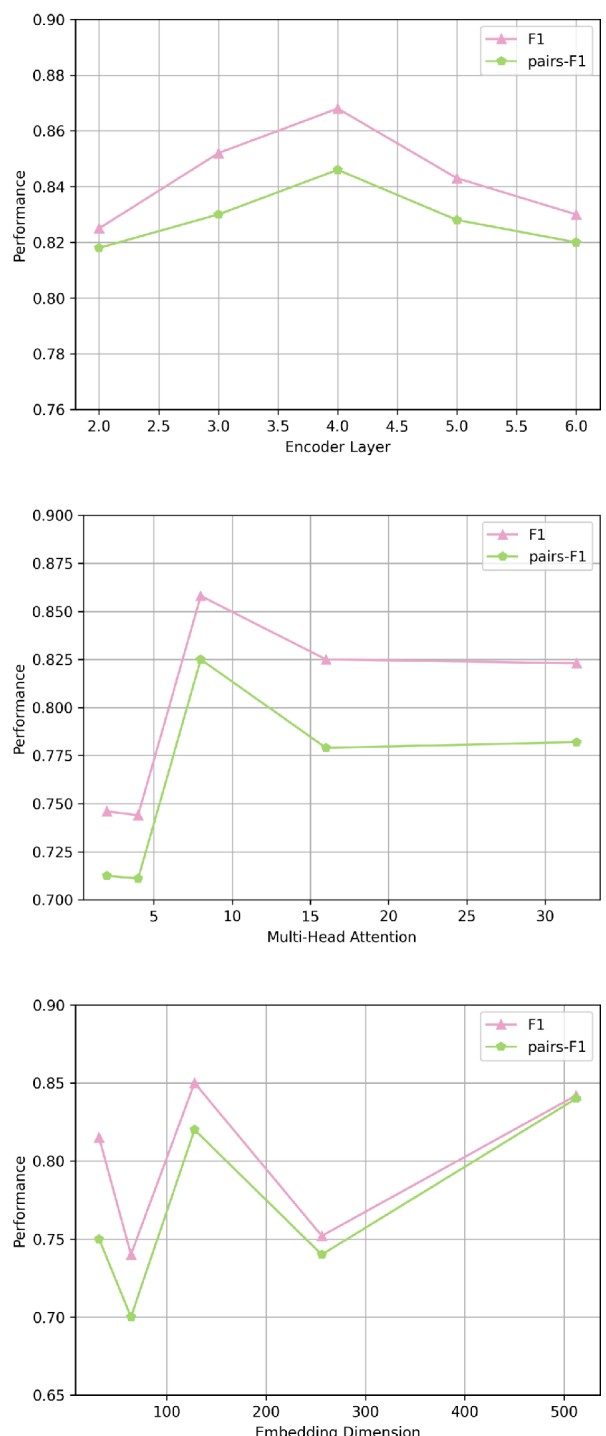

**Figure 7** **The parameter sensitivity analysis of Osaka.**

the complexity of tourist travel needs, we suggested several promising research directions in smart tourism. However, there are still some improvement directions for our proposed model. Firstly, is it possible to include other additional information such as social and weather information in the point-of-interest representation. In addition, on the issue of exploring user movement patterns, is there a more effective data augmentation strategy to capture user dynamics, such as exploring the possibility of migrating some strategies from the computer vision domain or the natural language processing domain.

### Funding

This work was supported by the National Natural Science Foundation of China under Grant 62072409. There was no additional external funding received for this study. The funders had no role in study design, data collection and analysis, decision to publish, or preparation of the manuscript.

### Grant Disclosures

The following grant information was disclosed by the authors:
The National Natural Science Foundation of China: 62072409.

### Competing Interests

Xiangjie Kong is an Academic Editor for PeerJ.

### Author Contributions

- Yuqi Shen conceived and designed the experiments, performed the experiments, analyzed the data, performed the computation work, prepared figures and/or tables, authored or reviewed drafts of the article, and approved the final draft.
- Yuhan Wu conceived and designed the experiments, performed the experiments, analyzed the data, performed the computation work, prepared figures and/or tables, authored or reviewed drafts of the article, and approved the final draft.
- Jingbo Song conceived and designed the experiments, analyzed the data, performed the computation work, authored or reviewed drafts of the article, and approved the final draft.
- Xiangjie Kong conceived and designed the experiments, performed the computation work, authored or reviewed drafts of the article, and approved the final draft.
- Giovanni Pau conceived and designed the experiments, analyzed the data, performed the computation work, authored or reviewed drafts of the article, and approved the final draft.

### Data Availability

The raw data and code are available in the Supplementary Files. The dataset is also available at Yahoo Datasets: https://webscope.sandbox.yahoo.com/catalog.php?datatype=i&did=67.

## Supplemental Information

Supplemental information for this article can be found online at http://dx.doi.org/10.7717/peerj-cs.2375#supplemental-information.

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
