# Peer review of "Enabling personalized smart tourism with location-based social networks"

_PeerJ Computer Science, doi:10.7717/peerj-cs.2375_

## Round 0.1 · original submission · Major Revisions

Please carefully revise your paper according to experts comments and mine below before resubmit.

Explain the relevance of LBSNs in more detail. How are they currently used in tourism, and what specific challenges do they face?

The explanation of how representation learning and contrastive learning are used could be more detailed. Include a brief overview of these techniques and how they specifically address the challenges mentioned.

Describe the specific architecture or model used in the proposed method.
Discuss any limitations or areas for further research identified through the case study.

Include examples or case scenarios where understanding human mobility patterns led to better recommendations or enhanced tourist experiences

Ensure all technical terms are defined and explained, especially for a diverse audience that may not be familiar with deep learning or LBSNs. This will make the paper more accessible

·

Basic reporting

Clarity and Language:
The manuscript is well-written in clear and professional English. The language used is appropriate for an academic audience, and the terminology is consistent throughout the paper. However, there are minor grammatical errors and awkward phrasings that could be improved for better clarity. For example, in the abstract, "to thoroughly release the potential of LBSNs by deep learning technologies" could be revised to "to fully harness the potential of LBSNs using deep learning technologies."

Introduction and Background:
The introduction provides a comprehensive background and clearly explains the relevance of the study within the context of smart tourism. The literature review is thorough, referencing relevant and recent studies that contextualize the current research. The authors effectively identify the knowledge gap that their research aims to fill.

Structure:
The manuscript conforms to the standard structure expected in academic papers, including sections for the introduction, related work, methods, results, discussion, and conclusion. The structure facilitates ease of reading and comprehension.

Figures and Tables:
Figures and tables are relevant, high-quality, and well-labeled. They effectively complement the text, providing visual support for the data and results presented. However, ensure that all figures and tables are appropriately cited in the text.

Raw Data:
The raw data is supplied in accordance with PeerJ's policy, which supports the transparency and reproducibility of the research.

Experimental design

Original Research:
The study presents original primary research within the scope of the journal. The research question is well-defined, relevant, and meaningful. The authors clearly state how their research addresses an identified knowledge gap in the literature on smart tourism and LBSNs.

Methodology:
The methodology is described in sufficient detail to allow replication. The authors use a robust experimental design, incorporating both representation learning and contrastive learning techniques. However, the explanation of some methods could be elaborated for better understanding. For example, the process of how the contextual information from LBSNs is embedded could be detailed further.

Ethical Standards:
The investigation is performed to a high technical and ethical standard. There is no indication of ethical concerns in the data collection or analysis procedures.

Validity of the findings

Data and Analysis:
The data provided are robust, statistically sound, and well-controlled. The authors employ rigorous statistical methods to analyze the data, ensuring that the results are reliable.

Conclusions:
The conclusions are well-stated and directly linked to the original research question. They are supported by the results presented, providing a clear answer to the research question. The authors successfully limit their conclusions to what their data can support, avoiding overgeneralization.

Additional comments

Strengths:
- The integration of LBSNs with deep learning techniques for personalized smart tourism is innovative and addresses a significant gap in the literature.
- The study uses a comprehensive and methodologically sound approach, enhancing the reliability of the findings.
- The manuscript is well-structured and clearly written, facilitating understanding.

Weaknesses:
- Minor grammatical errors and awkward phrasings should be addressed to improve clarity.
- The explanation of some methodological aspects could be expanded for better comprehensibility.
- Ensure all figures and tables are correctly cited in the text.

Reviewer 2 ·

Basic reporting

All comments have been added in detail to the last section.

Experimental design

All comments have been added in detail to the last section.

Validity of the findings

All comments have been added in detail to the last section.

Additional comments

Review Report for PeerJ Computer Science
(Enabling personalized smart tourism with Location-Based Social networks)

1. Within the scope of the study, location-based social networks were analyzed with deep learning using open source dataset.

2. In the introduction section, components of smart tourism, the purpose of the study and its main contributions to the literature were clearly mentioned.

3. In the related works section, the literature within the scope of the subject was examined in terms of points of interest recommendation and personalized smart tourism. This section definitely needs to be edited. In order to emphasize the place and importance of the study in the literature more clearly, it is suggested to add a literature table consisting of various important columns such as dataset, pros and cons, and results to this section.

4. When the methods used for the study are examined, the points of interest context information and framework in location-based social networks are explained in detail and the originality section is clearly stated.

5. The use of the preferred dataset, Yahoo Flickr Creative Commons 100 million dataset, and its use in Osaka and Glasgow seems appropriate in terms of type and amount of dataset within the scope of the study.

6. Although the evaluation metrics type and details are deemed sufficient, it should be explained in more detail how the values/types of hyperparameters such as optimizer, batch size and learning rate are determined during the training phase within the scope of the proposed model and whether different trials are performed. Also, comment on how the changes (increase/decrease) in these parameters will affect the result.

7. Performing data augmentation analysis in terms of dataset has increased the quality of the study. However, it is recommended that how the selected data augmentation types are selected, their place in the literature and their comparison be made in more detail by referring to the literature.

As a result, although the study is interesting in terms of the subject and has the potential to contribute to the literature, all the sections mentioned above should be explained step by step by examining them in detail.

Cite this review as

---

## Round 0.2 · accepted · Accept

Thank you for revising the paper according to the comments of the experts. I am pleased to inform you that we are now happy to accept your article.

Thank you for your fine contribution

·

Basic reporting

The authors improved the manuscript.

Experimental design

The authors improved the manuscript.

Validity of the findings

The authors improved the manuscript.

Reviewer 2 ·

Basic reporting

All comments have been added in detail to the last section.

Experimental design

All comments have been added in detail to the last section.

Validity of the findings

All comments have been added in detail to the last section.

Additional comments

Review Report for PeerJ Computer Science
(Enabling personalized smart tourism with Location-Based Social networks)

Thanks for the revision. The reviewer comments have been answered in great detail and it is observed that changes have been made in the relevant places in the paper. Therefore, I recommend that the paper be accepted. I wish the authors success in their future papers.

Cite this review as